# Catalytic asymmetric constructions of nitrogen, boron and carbon continuous stereogenic centers

Guan Zhang[1,2], Junyi Jia[2], Xuzhao Du[3], Bofan Feng[2], Kai Yang [2], Peiyuan Yu [3] & Qiuling Song [2,4,5] ✉

Precise construction of molecular chirality is a longstanding scientific challenge in synthetic chemistry. Although methods to enantioselectively assemble consecutive carbon stereocenters are abundant, courses to establish enriched continuous heteroatomic stereocenters have acquired less attention and remain a tremendous challenge. Of those atoms in main group elements, nitrogen and boron are the most intractable ones to control their chirality. The copper-catalyzed asymmetric insertion reaction is one of the most significant transformations to construct C-B bonds in organic synthesis. Herein, we report a protocol to synthesize troublesome continuous heteroatomic stereocenters from cyclic amine boranes and diazo compounds via asymmetric B-H insertion reaction. The protocol is compatible with a wide range of amine boranes and diazo compounds, exhibiting high diastereo- and enantioselectivities. Mechanistic studies reveal a meaningful kinetic resolution pathway involved in the transformation and DFT calculations elucidate the origins of stereoselectivity and diastereoselectivity.

Chiral molecules or optically pure compounds play crucial roles in various fields[1–3], such as in organic synthesis[4], pharmaceuticals[5], biomedical sciences[6], and functional materials[7–9]. There are great differences between the two enantiomers in terms of physiological activity and photoelectric properties[10]. Owing to the unique properties and wide applications, chirality has been a subject of widespread scientific interest. The development of catalytic asymmetric synthetic strategies to approach chiral compounds has been a vital recent advance in organic chemistry[11]. With the advancement of modern synthetic chemistry, research in asymmetric synthesis has expanded be-yond carbon central chirality and now encompasses other heteroatom central chirality as well as the exploration of single and multiple contiguous chiral centers[12–21]. The reports for the construction of contiguous carbon stereocenters or vicinal carbon-carbon stereogenic centers are relatively abundant and have been well developed[22–25], while the building of vicinal carbon-heteroatom stereogenic centers is still in its infancy[13,15], moreover, the construction of vicinal heteroatom-heteroatom or contiguous carbon-heteroatom-heteroatom stereocenters has not been reported yet (Fig. 1a).

In fact, in the family of heteroatomic stereocenters, the construction of nitrogen stereocenter is very challenging, since it tends to undergo rapid racemization due to a relatively low inversion barrier between the two N enantiomers (Fig. 1b)[26]. In this regard, chiral nitrogen-stereogenic centers often involve N-centered quaternary ammonium salts[27–29] or amine N-oxides[30–36], N-centered metal coordination[37,38], or rigid skeletons of tertiary amines[39–41]. Chiral nitrogen-stereogenic centers could be occasionally found in some natural products[42], drug molecules as well as chiral catalysts[30,31]. Since

[1]College of Chemistry and Materials Science, Fujian Normal University, Fuzhou, Fujian, China. [2]Key Laboratory of Molecule Synthesis and Function Discovery, Fujian Province University, College of Chemistry at Fuzhou University, Fuzhou, Fujian, China. [3]Department of Chemistry and Shenzhen Grubbs Institute, Southern University of Science and Technology Shenzhen, Guangdong, China. [4]State Key Laboratory of Coordination Chemistry, School of Chemistry and Chemical Engineering Nanjing University, Nanjing, China. [5]School of Chemistry and Chemical Engineering, Henan Normal University, Xinxiang, Henan, China. ✉e-mail: qsong@fzu.edu.cn

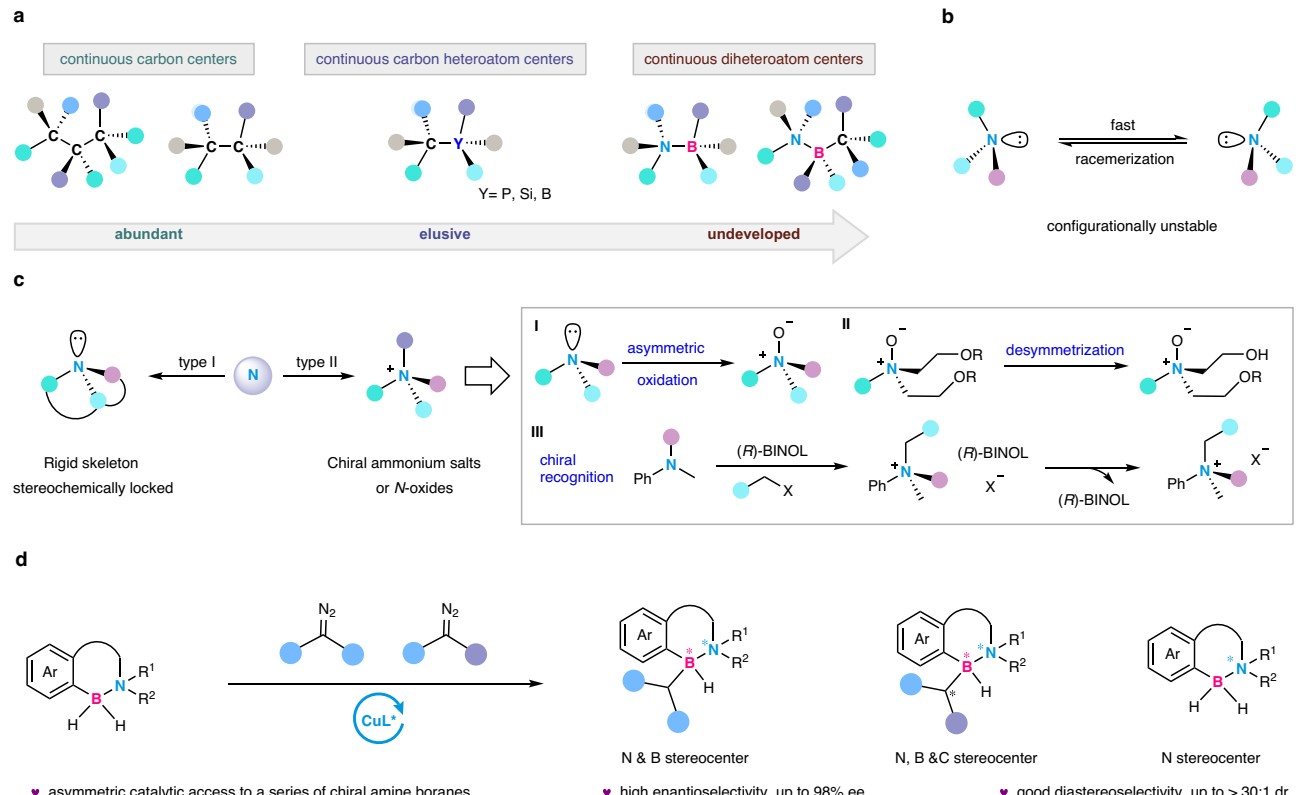

**Fig. 1 | Current status of the construction of contiguous (heteroatomic) stereocenters and N stereocenters and our strategy. a** Catalytic asymmetric construction of continuous stereocenters. **b** The challenge on nitrogen stereocenter. **c** The strategies for the synthesis of N stereocenters. **d** Catalytic asymmetric B-H insertion to access N, N and B, as well as N, B, C contiguous stereocenters. BINOL: 1,1′-Bi-2-naphthol.

the lone pair electron on nitrogen atom is locked, some chiral tertiary amines with fixed configurations are generated, as in the rigid skeleton of Quinine and its derivatives. Alternatively, limited asymmetric oxidation and desymmetry examples were applied to the construction of chiral *N*-oxides[32–36]. And the configuration of nitrogen atoms in quaternary ammonium salts were set by chiral BINOL as the chiral induced-reagents, of note, in which stoichiometric chiral reagents were needed to fulfill the goal (Fig. 1c)[27,28]. To date, accessing chiral compounds with nitrogen as the stereogenic element is still elusive and challenging, let alone the construction of chiral compounds with continuous heteroatomic stereocenters including one N atom. Since we were very interested in elemental chemistry, we envisioned that a variety of chiral compounds with contiguous heteroatomic stereogenic centers can be obtained by one-step conversion of racemic amine boranes, since the empty P orbital of boron atom could accommodate the lone pair electron of nitrogen atom, which is the key to control the configuration of nitrogen. Enantioselective B-H bond insertion has emerged as an important strategy for constructing chiral compounds bearing carbon or boron stereocenters.[20,21,43–54]. Herein, we report a copper-catalyzed asymmetric B-H insertion reaction between racemic amine boranes and diaryl diazomethanes, enabling the construction of both vicinal nitrogen & boron stereogenic centers and nitrogen, boron & carbon continuous stereogenic frameworks (Fig. 1d). Moreover, through a kinetic resolution process, enantiopure boron-coordinated nitrogen-centered compounds can be prepared, which can serve as potential chiral transfer hydrogenation reagents for the enantioselective reduction of ketones and imines.

## Results and discussion

To validate our hypothesis, we chose amine borane **1a** and symmetric diazo compound **2a** as model substrates and subjected them to a copper catalysis. Based on our previous experience, we recognized that the ability to catalytically activate B-H bonds hinging on the activity of chiral BOX (bisoxazoline) ligands. Therefore, the reaction was firstly examined in DCM (dichloromethane) as solvent at −70 °C with the combination of various BOX ligands and CuTc (copper(I) thiophene-2-carboxylate hydrate) as catalysts, and the results were listed in Table 1. Among the evaluated six ligands (**L1**–**L6**), ligands **L1** and **L6** were uncommercialized ones, but they exhibited better stereoselectivities compared to the others (entries 1–6). When **L6** was used, a diazo homo-coupling byproduct was obtained thus leading to a lower yield for the desired product **3a**, ligand **L1** was outstanding and superior both in efficiency and stereoselectivity (entries 1 and 6). When the additive KBAr$_F$ (potassium tetrakis(perfluorophenyl)borate) was absent or added with reduced loading amount, the yields of desired product **3a** were reduced as well, moreover, when NaBAr$_F$ was used to replace KBAr$_F$, the efficiency and enantioselectivity were both suppressed, indicating that the additive KBAr$_F$ was important to increase the reaction efficiency (entries 7–9). We found that increasing temperature also decreased enantioselectivity (entries 10 and 11). When Cu(MeCN)$_4$PF$_6$ was utilized as catalyst to take place of CuTc in DCM as solvent at −70 °C with the combination of **L1** (20 mol%) in the absence of additive, a good yield and enantioselectivity for the desired product **3a** was also obtained (entry 12), although it is a little bit inferior to the condition in entry 1 of Table 1.

Through condition optimization, we identified two optimal copper metals with the same ligand **L1** (entries 1 and 12), among them the additive was required when CuTc was used. With the optimized ligand and catalysts in hand, we sought to explore the generality of the asymmetric hydrogen transfer protocol first with various diazo compounds (Fig. 2, top). For the insertion reactions of symmetric diazo compounds **2** with amine borane **1a**, good to excellent yields and high

**Table 1 | Optimization of the reaction conditions**

| Entry | Ligand | Additive | Temperature (°C) | Yield (%)[b] | ee (%)[c] | dr[d] |
|---|---|---|---|---|---|---|
| 1[a] | L1 | KBAr$_F$ | −70 | 90 | 91 | >20:1 |
| 2 | L2 | KBAr$_F$ | −70 | 75 | 42 | >20:1 |
| 3 | L3 | KBAr$_F$ | −70 | 78 | 68 | >20:1 |
| 4 | L4 | KBAr$_F$ | −70 | 88 | 79 | >20:1 |
| 5 | L5 | KBAr$_F$ | −70 | 70 | −48 | >20:1 |
| 6 | L6 | KBAr$_F$ | −70 | 25 | 90 | >20:1 |
| 7 | L1 | No | −70 | 56 | 85 | >20:1 |
| 8[e] | L1 | KBAr$_F$ | −70 | 78 | 87 | >20:1 |
| 9 | L1 | KBAr$_F$ | −70 | 51 | 88 | >20:1 |
| 10 | L1 | KBAr$_F$ | −40 | 91 | 77 | >20:1 |
| 11 | L1 | KBAr$_F$ | −60 | 90 | 85 | >20:1 |
| 12[f] | L1 | No | −70 | 88 | 89 | >20:1 |

[a] Conditions: amine borane **1a** (0.3 mmol, 3.0 equiv), (diazomethylene)dibenzene (**2a**) (0.1 mmol, 1.0 equiv), CuTc (10 mol%), **L** (11 mol%), KBAr$_F$ (10 mol%), DCM (2 mL) at −70 °C, argon, 48 h.
[b] Isolated yield.
[c] Enantiomeric excess (ee) was determined by chiral HPLC.
[d] Diastereomeric ratio (dr) was determined by chiral HPLC and crude [1]H NMR.
[e] KBAr$_F$ (5 mol%).
[f] Cu(MeCN)$_4$PF$_6$ instead of CuTc, **L** (20 mol%), and no additive.

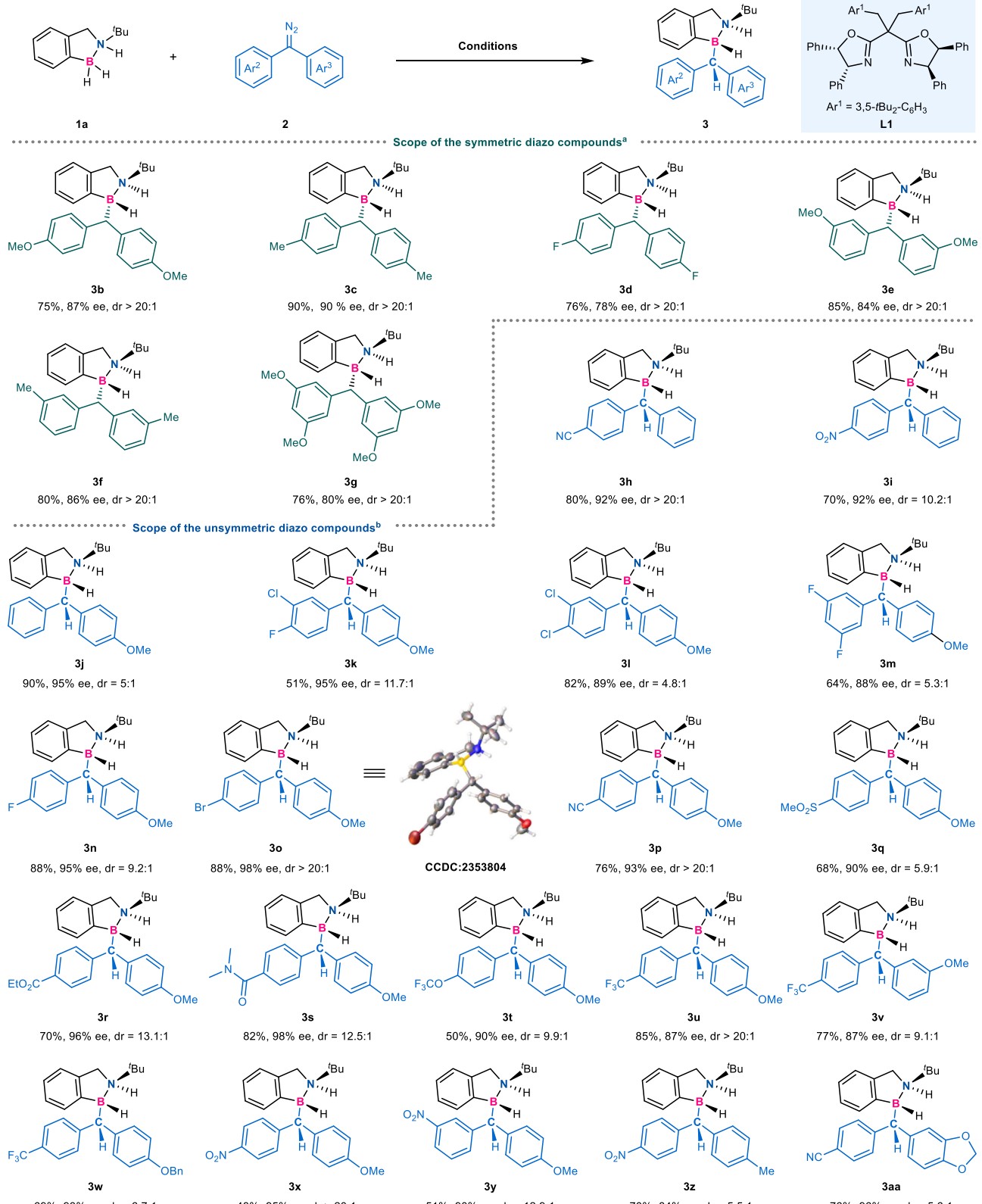

**Fig. 2 | Scope of the diazo compounds.** Conditions: [a] amine borane **1a** (0.3 mmol, 3.0 equiv), diaryl diazomethanes **2** (0.1 mmol, 1.0 equiv), CuTc (10 mol%), **L1** (11 mol%), KBAr$_F$ (10 mol%), DCM (2 mL) at −70 °C, argon, 48 h, isolated yield. Ee and dr

were determined by chiral HPLC. [b] amine borane **1a** (0.3 mmol, 3.0 equiv), diaryl diazomethanes **2** (0.1 mmol, 1.0 equiv), Cu(MeCN)$_4$PF$_6$ (10 mol%), **L1** (20 mol%), DCM (2 mL) at −70 °C, argon, 48 h, isolated yield.

enantioselectivities were obtained with -OMe (**2b**), -Me (**2c**) and -F (**2d**) substituents in the para-position of diaryl diazomethanes (**3b**–**3d**). The meta-substituted substrates (**2e** and **2f**) also had good compatibility (**3e**–**3f**), but for the disubstituted alkoxy substrate (**2g**), the

enantioselectivity was reduced (**3g**). Encouraged by the results, we next turned our attention to the scope of unsymmetric diaryl diazomethanes with amine borane **1a** (Fig. 2, bottom). For example, para-substituents on the aryl ring of diazo compounds, including -CN, -NO₂,

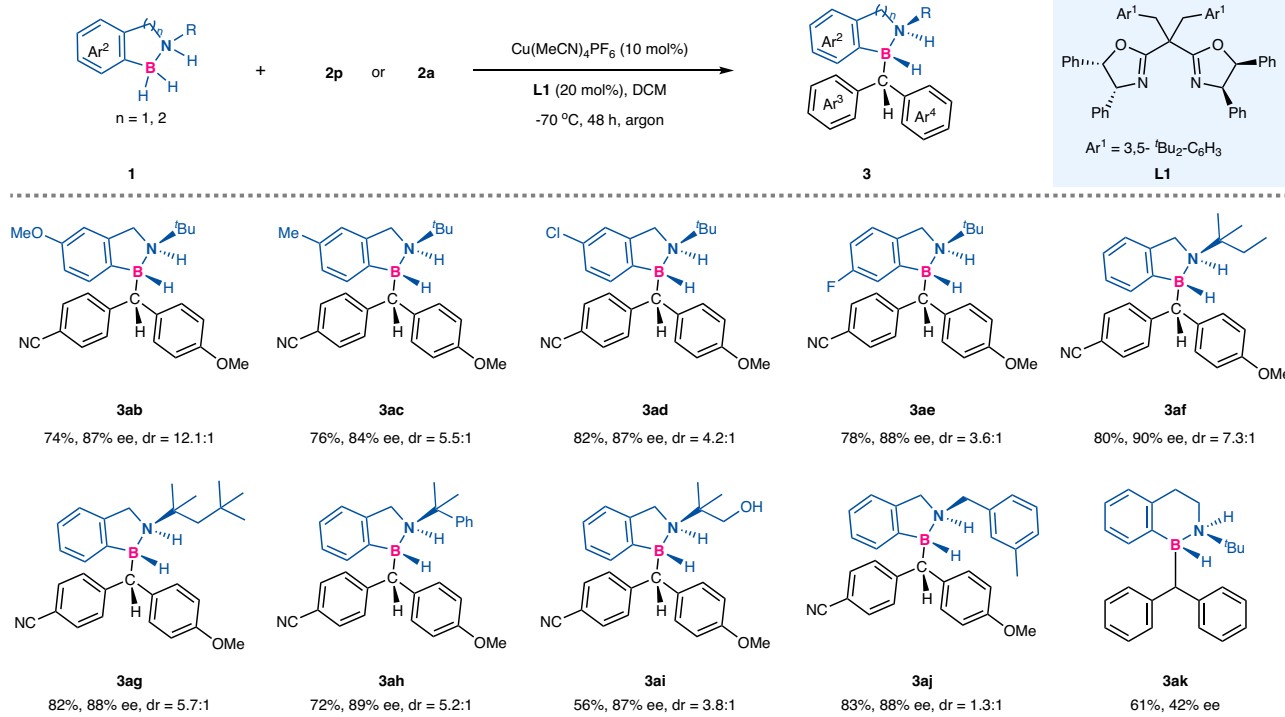

**Fig. 3 | Scope of the amine boranes.** Conditions: amine boranes **1** (0.3 mmol, 3.0 equiv), diaryl diazomethanes **2** (0.1 mmol, 1.0 equiv), Cu(MeCN)$_4$PF$_6$ (10 mol%), **L1** (20 mol%), DCM (2 mL) at −70 °C argon, 48 h, isolated yield. Ee and dr were determined by chiral HPLC.

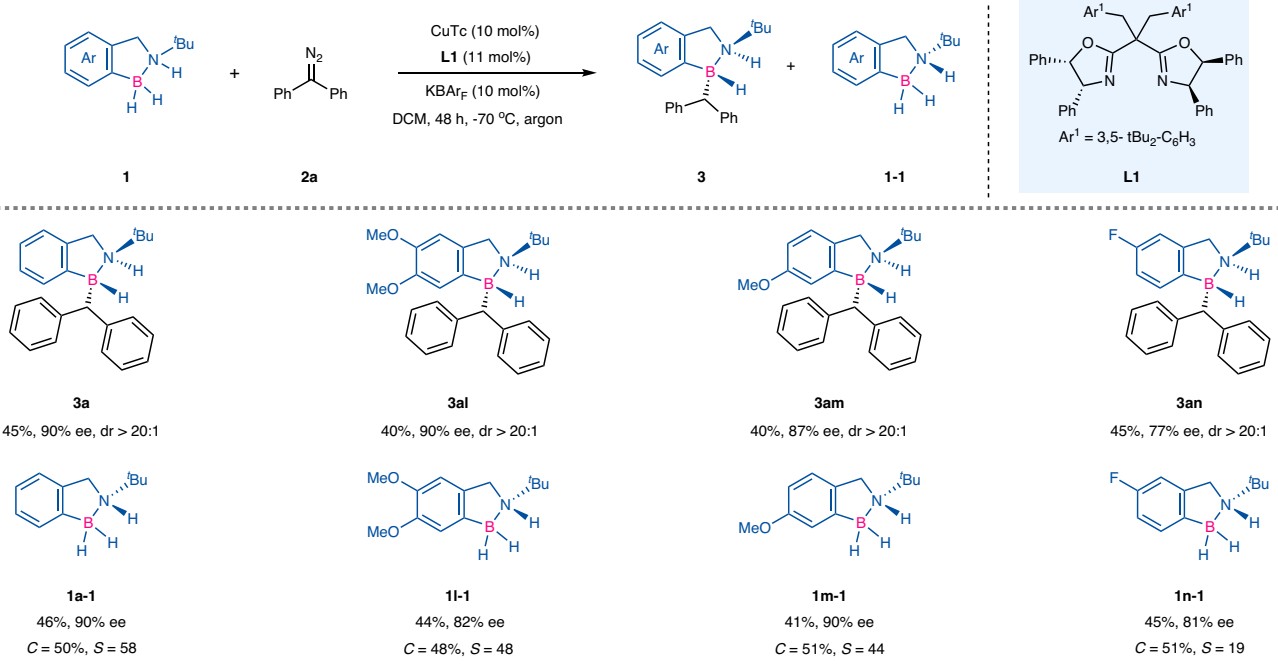

**Fig. 4 | Kinetic resolution of this Cu-catalyzed B-H insertion reaction for racemic amine boranes.** Conditions: amine boranes **1** (0.2 mmol, 2.0 equiv), (diazomethylene)dibenzene (**2a**) (0.1 mmol, 1.0 equiv), CuTc (10 mol%), **L1** (11 mol%), KBAr$_F$ (10 mol%), DCM (2 mL) at −70 °C, argon, 48 h, isolated yield. Ee and dr were determined by chiral HPLC.

-OMe, were all well tolerated, furnishing **3h, 3i, 3j** in 70–90% yields with 92–95% ee, and among them, electron-withdrawing groups had better diastereoselectivities. When two different types of substituents, such as electron-donating groups (OMe, OBn, Me, diether) and electron-withdrawing groups (F, Cl, Br, CN, SO$_2$Me, COOEt, CONMe$_2$, OCF$_3$, CF$_3$ and NO$_2$) were on Ar$^2$ and Ar$^3$ rings respectively, the substrates all demonstrated good reactivities and the corresponding products (**3k**–**3aa**) were obtained in good yields and high enantioselectivities (48–88% yields with 84–98% ee, 4.8:1 – >20:1 dr values). Of note, when Ar$^2$ had -CF$_3$ group, substrates with para- or

**a**

**b**

| entry | 1a:2a | 3a ee (%) | 3a Yield (%) | 1a-1 ee (%) | 1a-1 Yield (%) |
|---|---|---|---|---|---|
| 1 | 1:0.6 | 69 | 60 | 94 | 32 |
| 2 | 1:0.55 | 83 | 49 | 92 | 47 |
| 3 | 1:0.5 | 90 | 46 | 90 | 49 |
| 4 | 1:0.47 | 90 | 45 | 81 | 54 |
| 5 | 1:0.45 | 91 | 40 | 73 | 53 |

**c**

**d**

**e**

**Fig. 5 | Mechanistic studies and transformations of the amine boranes. a** The enantioselectivities of **3a** and **1a–1** versus time. **b** The effect of the ratio of **1a** to **2a** on the reaction. **c** The deuterium-labeling experiment. **d** Ketone and amine reduction by chiral amine borane. **e** Synthetic transformations. Yields are isolated. Ee was determined by chiral HPLC.

meta-substituted Ar³ rings also showed fairly good enantioselectivities (**3u** and **3v**), but para-substituted Ar³ had a better diastereoselectivity than meta-substituted one. Slightly decreased diastereoselectivities were obtained when other alkoxy groups (**3w** and **3aa**) were investigated. When diazo compounds with Ar² ring bearing an electron-withdrawing para-nitro group, the stereoselectivity decreased as the electron-donating ability of the substituents on Ar³ ring decreased (**3x** vs **3z, 3p** vs **3h, 3x** vs **3i**). On the contrary, when diazo compounds with Ar³ ring containing an electron-donating para-methoxy group, the stereoselectivity decreased as the electron-withdrawing ability of the substituents on Ar² ring decreased (**3t** vs **3u**). The stereoselectivity was affected by the electronic differences of the substituents on both Ar² and Ar³ of diaryl diazomethanes.

Moreover, this strategy proved to be suitable to other substituted amine boranes as well. As shown in Fig. 3, when OMe, Me, Cl and F

groups were introduced into the Ar² ring, the substrates were all good candidates for this insertion reaction and good to excellent yields and enantioselectivities were smoothly obtained for contiguous carbon–diheteroatom stereocenters (**3ab**–**3ae**), albeit in slightly lower diastereoselectivities. When R substituent instead of ᵗBu group on nitrogen atom were used, not surprisingly, contiguous carbon–diheteroatom stereocenters were observed with good yields and enantioselectivities with diastereoselectivity enhanced (**3af**–**3aj**). It is worth mentioning that the substrate containing free hydroxyl group was also compatible to the standard conditions, and no O-H bond insertion was found (**3ai**). Furthermore, sixmembered cyclic amine borane compound ($n = 2$) was suitable for this insertion reaction as well to deliver the corresponding target product **3ak**, a lower enantioselectivity was observed probably due to the fact that the ligand was not the optimal ligand since no further optimization was performed on it.

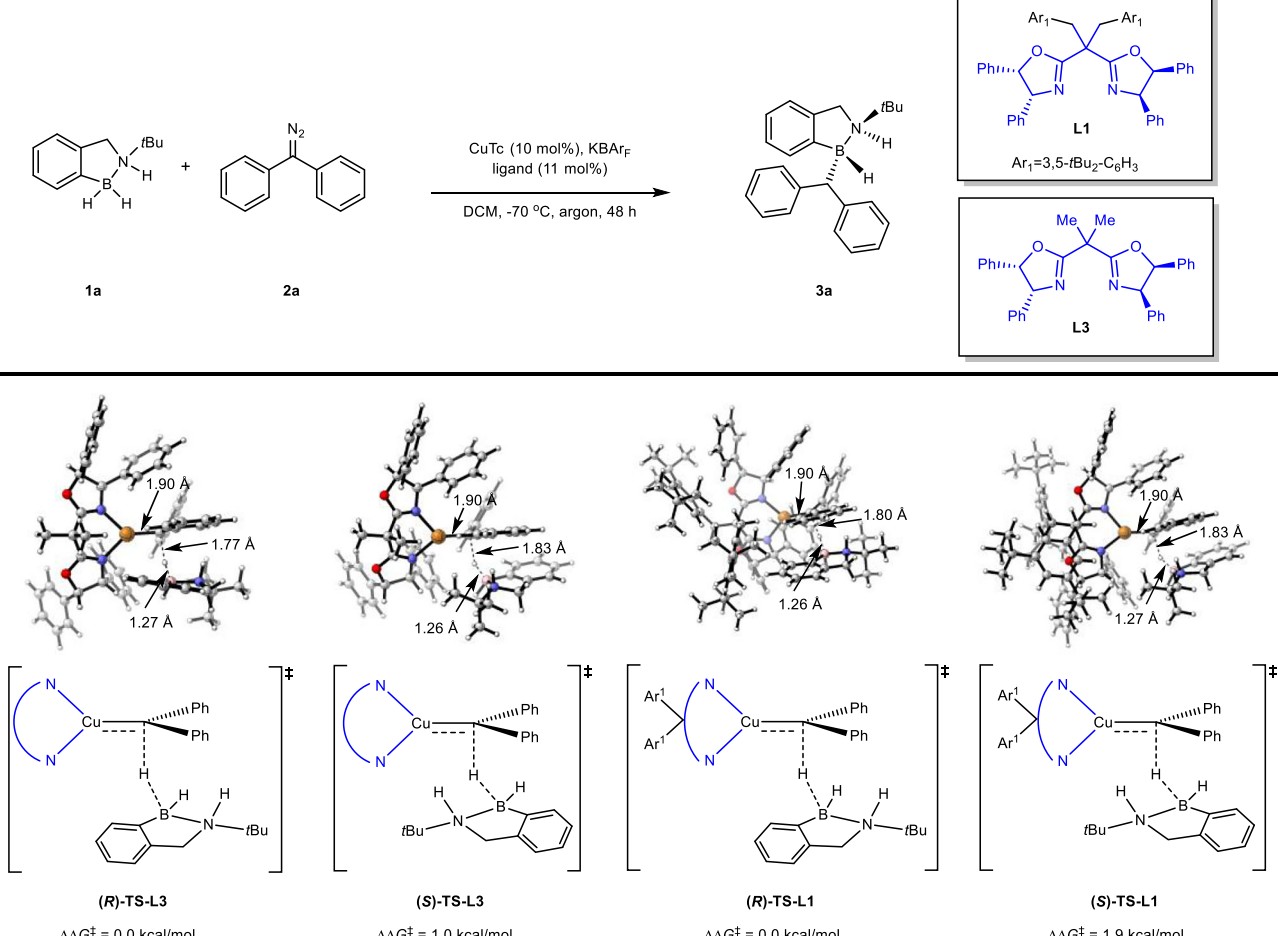

**Fig. 6 | DFT-calculated key transition state structures for the enantio-determining hydride transfer step with 1a and 2a as substrates.** Calculations were performed at PBE0-D3(BJ)/def2-TZVP/SMD(DCM)//B3LYP-D3(BJ)/def2-SVP/SMD(DCM) level of theory.

On the basis of our observations, we speculated that the reaction might undergo a kinetic resolution for racemic amine boranes. With the optimal conditions accessible, if the amount of amine borane **1** is reduced to make the ratio of it to diazo compound **2** as 2:1, a kinetic resolution of amine borane **2** should be realized. This strategy resulted in the compounds with only one nitrogen stereogenic center (the unreactive starting material), as well as the compounds with contiguous heteroatomic stereogenic centers (Fig. 4). Model substrate **1a** was used to provide the product **3a** with both an exciting enantioselectivity and yield, and the recovered chiral amine borane (**1a−1**) also had good enantioselectivity (90% ee) with excellent recovery rate (46%). The kinetic resolution of **rac-1l** was completed, providing the desired product **3al** in 40% yield with 90% ee and the recovered chiral amine borane **1l−1** in 44% yield with 82% ee. When mono substituted substrates (with MeO or F groups) were used, the recovered chiral amine boranes were obtained in good enantioselectivities (**1m−1** and **1n−1**).

To unveil the reaction process of the kinetic resolution, the following experiments were conducted. First, when the amount of amine borane **1a** and diazo compound **2a** was 2:1, the profile of the enantioselectivity of the recovered **1a−1** and the product **3a** versus time indicated that the reaction was process of kinetic resolution (Fig. 5a, for details, please see the supporting information for the specific experiment process). Then, in order to understand the process of kinetic resolution, the effect of **1a** and **2a** ratio on ee values and yields of **3a** and **1a−1** was further illustrated (Fig. 5b). When the amount of **1a** was constant and **2a** changed from large to small, the ee value of **3a**

was increased, and the ee value of **1a−1** was decreased (Fig. 5b). The above experiments suggested that racemic amine boranes underwent a process of kinetic resolution. To acquire some insights into the pathway of hydrogen-transfer process, deuterated amine borane (**1a-d₃**) was prepared. One of the deuterium atoms was moved from boron to the α-carbon of the boron/nitrogen vicinal stereocenters to deliver product **3a-d₃** under the standard conditions (Fig. 5c). The chiral amine borane **1a−1** was obtained by kinetic resolution, which could be used as a chiral transfer hydrogenation reagent[55,56] to reduce ketone and amines (Fig. 5d). Although the enantioselectivity was not high, it indicated that this would be a useful and valuable type of chiral amine boranes with a chiral N stereocenter, which have not been reported before. The product **3o** with three continuous stereogenic centers could undergo Negishi coupling for further arylation and alkylation (**6** and **7**) with high retention of the enantiopurity (Fig. 5e).

To explain the origin of stereoselectivity in the reaction and the excellent selectivity of ligand **L1**, we summarized the common structural features of ligands **L1**–**L6**, constructed a model ligand **L3**, and studied the asymmetric hydrogen transfer steps of both **L3** and **L1** in the reaction shown at the top of Fig. 6 through DFT calculations. The corresponding pair of transition states, (**R**)-**TS-L3** and (**S**)-**TS-L3**, are shown in the bottom of Fig. 6. Energy calculations indicate that the pathway through (**R**)-**TS-L3** has a lower activation free energy barrier, with (**S**)-**TS-L3** being 1.0 kcal/mol higher (at 203.15 K). Furthermore, we investigated both transition states through IGMH analysis to visualize how non-covalent interactions influence reaction selectivity, as illustrated in Fig. 7. The IGMH analysis results show that one of the

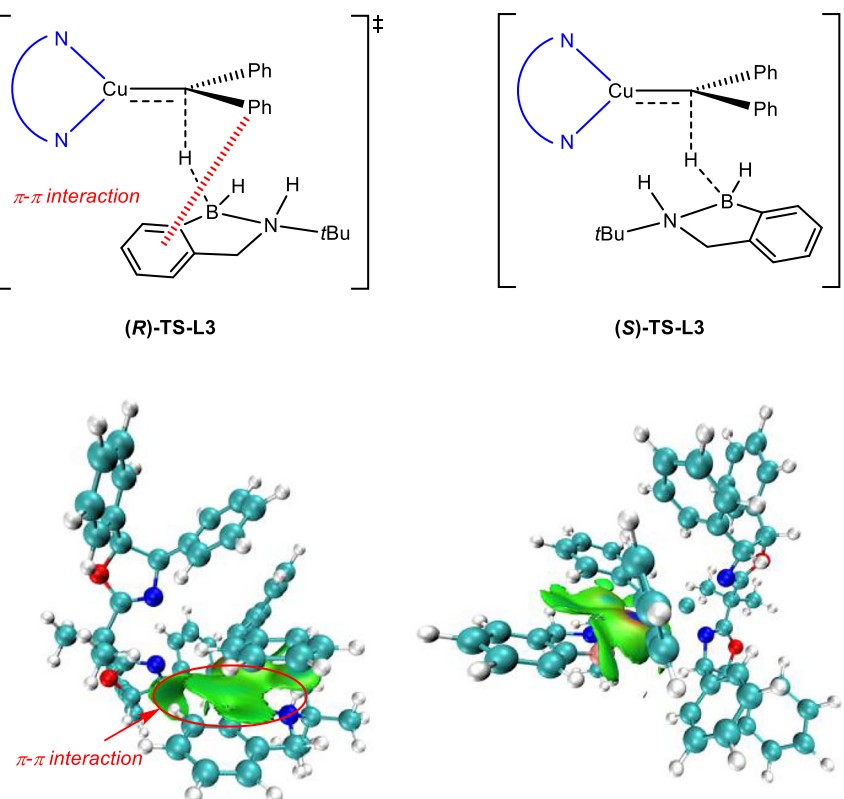

**Fig. 7** | The IGMH analysis for the transition states (*R*)-TS-L3 and (*S*)-TS-L3.

aromatic groups of the symmetric diazo compound **2a** in (*R*)-**TS-L3** effectively overlaps with the aromatic group of amine borane **1a**, forming distinct π-π interaction, which we postulate as one source of the reaction's stereoselectivity.

To better quantify the stereoselectivity factors in the transition states, we conducted Distortion-Interaction analysis on both transition states involving the **L3** ligand, with results presented in Supplementary Table S7 in Supporting Information (SI). The analysis reveals that while the (*R*)-**TS-L3** transition state structure shows a slight disadvantage in distortion energy (+0.3 kcal/mol) compared to (*S*)-**TS-L3**. It demonstrates a significant advantage in interaction energy (−0.7 kcal/mol). This DI analysis further corroborates the results of IGMH analysis, confirming that π-π interaction between substrates is crucial in favoring *R*-configured product formation. Subsequently, building upon the **L3** ligand catalytic model, we further investigated the reaction transition states involving ligand **L1** to provide a rational explanation for its relatively higher stereoselectivity and guide future ligand optimization. As shown in Fig. 6, the asymmetric hydrogen transfer transition states for ligand **L1** are (*R*)-**TS-L1** and (*S*)-**TS-L1**, with a calculated energy difference of 1.9 kcal/mol, slightly higher than the corresponding experimental value of 1.3 kcal/mol (91% *ee* at 203 K). Considering that the main structural difference between **L1** and **L3** lies in the two outward-facing aromatic substituents, we reasoned that their effect on π-π interaction energy would be minimal. Therefore, we directly employed DI analysis for further investigation, with results shown in Supplementary Table S7 in SI. The calculations reveal that, compared to transition states involving ligand **L3**, the main difference lies in the significantly increased distortion of the diazo compound in (*S*)-**TS-L1** relative to (*R*)-**TS-L1**. This results in the *R*-configured transition state of **L1**-catalyzed hydride transfer having both substantial advantages in interaction energy (−1.6 kcal/mol) and distortion energy (−1.5 kcal/mol) (for more details, please see Supplementary Table S7 in SI).

Comparing the transition state structures and DI analysis results of **L3** and **L1** ligands, we propose that the weak interactions between

the aromatic group in the symmetric diazo compound and the aromatic group in amine borane are one of the key factors leading to stereoselectivity tendency in the reaction. Meanwhile, the excellent selectivity of ligand **L1** may be due to the large steric hindrance between its bulky substituents and the substituent in amine borane, which leads to further distortion of the copper carbene fragment in the transition state, thereby increasing the activation energy difference between transition states.

In summary, we have presented the Cu-catalyzed asymmetric B-H bond insertion reactions assembling contiguous three atomic (B, N, C) stereocenters, vicinal two heteroatomic (B and N) stereocenters and nitrogen stereocenter from readily accessible amine boranes and diazo compounds. It represents a successful construction of three consecutive chiral centers involving different types of atoms, thereby expanding the diversity of chiral compounds. Mechanistic experiments demonstrated it via a kinetic resolution process and DFT calculations explained the origin of stereoselectivity and diastereoselectivity.

## Methods

### General procedure for the synthesis of chiral amine boranes
In air, a 25 mL Schlenk tube was charged with **1**, **2**, copper metals, ligand and additive. The tube was evacuated and filled with argon for three cycles. Then, 1 mL of dichloromethane was added under argon. The reaction was allowed to stir at −70 °C for 48 h. Upon completion, proper amount of silica gel was added to the reaction mixture. After removal of the solvent, the crude reaction mixture was purified on silica gel (petroleum ether and ethyl acetate) to afford the desired amine borane products.

## Data availability
The data that support the findings of this study are available within the article and its Supplementary Information files. All data are available from the corresponding author upon request. The X-ray

crystallographic coordinates for structures reported in this study have been deposited at the Cambridge Crystallographic Data Centre (CCDC), under deposition number 2353804 (**3o**). These data can be obtained free of charge from The Cambridge Crystallographic Data Centre via www.ccdc.cam.ac.uk/data_request/cif. Source Data are provided with this manuscript. Source data are provided with this paper.

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

## Acknowledgements

Financial support from National Key R&D Program of China (2023YFF0723900 to Q.S.), National Natural Science Foundation of China (22271105, 21931013 to Q.S.), Natural Science Foundation of Fujian Province (2022J05016 to Q.S.), Open Research Fund of State Key Laboratory of Coordination Chemistry, School of Chemistry and Chemical Engineering, Nanjing University and Open Research Fund of School of Chemistry and Chemical Engineering, Henan Normal University are gratefully acknowledged. Computational work was supported by the Center for Computational Science and Engineering and the CHEM High-Performance Supercomputer Cluster (CHEM-HPC) of the Department of Chemistry, Southern University of Science and Technology

## Author contributions

Q.S. conceived and directed the project. G.Z. and J.J. performed experiments and prepared the supplementary information. B.F. and K.Y. helped collecting some new compounds analyzing the data. P.Y. and X.D. performed the DFT calculations and drafted the DFT parts. Q.S. and G.Z. wrote the paper. All authors discussed the results and commented on the manuscript.

## Competing interests

The authors declare no competing interests.
