## [Transparent Peer Review file · Nature Communications]

Catalytic asymmetric constructions of nitrogen, boron and carbon continuous stereogenic centers

Corresponding Author: Professor Qiuling Song

Version 0:

Reviewer comments:

Reviewer #1

(Remarks to the Author)

The manuscript by Song and co-workers describes a Cu-catalyzed asymmetric insertion reaction of cyclic amine-boranes with diazo compounds involving a kinetic resolution process. This protocol provides an efficient approach to synthesizing a range of structures bearing continuous contiguous carbon-boron-nitrogen or boron-nitrogen stereocenters or nitrogen stereocenter, while sometimes the enantioselectivity and diastereoselectivity are not so satisfied. Although the construction of contiguous carbon stereocenters or vicinal carbon-carbon stereogenic centers are relatively abundant, the development of vicinal carbon-heteroatom stereogenic centers is still rare. The obtained chiral product was applied as chiral catalyst to asymmetric hydrogenation of ketones. Overall, this is an interesting report on the simultaneous construction of carbon- and heteroatom-centered chirality. Based on the novelty of this research, I recommend its publication in Nature Communications after the following revisions.

- (1) Regarding the compounds with two or three stereocenters reported in this work, the authors claimed that they observed a diastereoselectivity with a range from 1.3/1 to >30/1. It is easy to understand that the dr values are extremely high in compounds with boron-nitrogen stereocenters due to the efficient kinetic resolution, while it is hard to control the enantioselectivity of carbon stereocenter. However, the authors did not specify how the diastereomeric ratios were determined—whether by NMR, GC, or another method. Besides, could the diastereoisomers be separated by column chromatography? All of the above information should be clearly presented in the main text or Supporting Information.
- (2) The authors did not provide any optimization of the reaction conditions in the Supporting Information. Including detailed reaction conditions is essential to help readers better understand the development and reproducibility of the method.
- (3) In Table 1, the dr values should also be provided, and what is the structure of Ar₃ in ligand screening?
- (4) What is the origin of diastereoselectivity in the cases involving unsymmetrical diaryl diazomethanes? According to the DFT calculation results, it is likely that the $\pi \dots \pi$ interaction between the aromatic group in the symmetric diazo compound and the aromatic group in amine borane, is the key factor leading to stereoselectivity tendency in the reaction. Detailed DFT calculation is suggested to give a clearer explanation for the unsymmetrical diaryl diazomethanes.
- (5) In Figure 5b, the S factor and C values should also be provided.
- (6) In Figure 5d, 1a-1 was used as a chiral catalyst; however, the hydrogenated product was obtained in only 35% yield with 17% ee. Further optimization is clearly needed.

Reviewer #2

(Remarks to the Author)

Asymmetric synthesis of chiral molecules has long been a significant challenge in organic chemistry. Over the past few decades, numerous approaches have been explored for constructing C-centered chirality, while precise modulation of contiguous heteroatomic stereocenters, particularly those involving nitrogen and boron, remains complex. In this context, Song and coworkers have presented a robust protocol for constructing troublesome contiguous three-atomic (B, N, C) stereocenters, vicinal two-heteroatomic (B and N) stereocenters, and a nitrogen stereocenter via Cu/BOX-catalyzed asymmetric B–H insertion. A wide range of amine boranes and diazo compounds demonstrated high enantio- and diastereoselectivities.

Mechanistic investigations suggest that the reaction may undergo a kinetic resolution of racemic amine boranes, resulting in the simultaneous synthesis of both chiral unreacted substrates with a single nitrogen stereogenic center and compounds featuring contiguous heteroatomic stereogenic centers. The enantioriched starting materials also show potential as chiral

organocatalysts in the asymmetric hydrogenation of naphthyl ketones, albeit with inferior enantiocontrol. Additionally, DFT calculations provided further insights into the stereocontrol model. These results are intriguing, and I recommend publication after major revisions to address several deficiencies:

- 1) The B–H bond is retained in the enantioriched product. Is it possible for this compound to undergo Cu-catalyzed B–H insertion again, potentially assisted by Int-L1 with the opposite configuration or a racemic BOX ligand?
- 2) Could the enantioriched product be utilized in an enantioselective hydrogenation, as illustrated in Fig. 5d? Would this provide superior enantiocontrol?
- 3) There are several typographical errors in the manuscript that should be carefully re-checked. Here are some examples:
 - a) In the abstract, redundant hyphens have been included, such as in “proto-col,” “trouble-some,” and “path-way.”
 - b) On page 2, line 11, the “N” in “N-oxides” should be italicized.
 - c) On page 2, line 20, a redundant space appears in “carbon- boron-nitrogen.”
 - d) In Figure 5b, the first line “3a Yield (%)” appears awkwardly formatted.

Reviewer #3

(Remarks to the Author)

The authors disclose an asymmetric B–H insertion reaction of diaryl diazo compounds by reacting the diazo compounds with cyclic boranes harboring an adjacent amine group. While the reaction is interesting, I am not convinced that it has enough conceptual novelty and significance to merit publication in Nat. Comm. The obtained products are highly specialized and do not provide a general access to a wide variety of B, N containing compounds.

i) Authors have already reported a conceptually similar copper catalyzed B-H insertion reaction of alpha-diazo phosphonates (Chemical Science, 2024,15, 7130-7135). Furthermore, several catalytic B-H insertion reactions have been reported on other systems (reference 43-54).

ii) If the novelty of the current work is on generating chiral nitrogen center adjacent to chiral borane, then applications of obtained products to construct biologically/medicinally important chiral B- or N- containing compounds should have been given. In the absence of any application of borane group, it is not clear how the presence of chiral borane atom adjacent to nitrogen (Fig. 2-4) would allow construction of valuable chiral amines.

Version 1:

Reviewer comments:

Reviewer #1

(Remarks to the Author)

The authors have fully addressed all my previous concerns, and I recommend this manuscript for publication in Nature Communications without further revision.

Reviewer #2

(Remarks to the Author)

The authors have properly revised the manuscript and addressed the concerns raised by the reviewers. The publication is recommended.

Reviewer #3

(Remarks to the Author)

The authors have provided additional data (Fig. 5d) to demonstrate the utility of their new compounds. Also, the relative importance and novelty of this work in comparison to their previous work has been clarified. The revised manuscript could be accepted for publication after the authors add this clarification so that this work can be adequately placed in context to the prior art.

Point-by-Point Response to the reviewers' comments

COMMENTS

Reviewer #1:

The manuscript by Song and co-workers describes a Cu-catalyzed asymmetric insertion reaction of cyclic amine-boranes with diazo compounds involving a kinetic resolution process. This protocol provides an efficient approach to synthesizing a range of structures bearing continuous contiguous carbon-boron-nitrogen or boron-nitrogen stereocenters or nitrogen stereocenter, while sometimes the enantioselectivity and diastereoselectivity are not so satisfied. Although the construction of contiguous carbon stereocenters or vicinal carbon-carbon stereogenic centers are relatively abundant, the development of vicinal carbon-heteroatom stereogenic centers is still rare. The obtained chiral product was applied as chiral catalyst to asymmetric hydrogenation of ketones. Overall, this is an interesting report on the simultaneous construction of carbon- and heteroatom-centered chirality. Based on the novelty of this research, I recommend its publication in Nature Communications after the following revisions.

Response: We sincerely thank this reviewer for the favorable comments on our work, we really appreciate it.

(1) Regarding the compounds with two or three stereocenters reported in this work, the authors claimed that they observed a diastereoselectivity with a range from 1.3/1 to >30/1. It is easy to understand that the dr values are extremely high in compounds with boron-nitrogen stereocenters due to the efficient kinetic resolution, while it is hard to control the enantioselectivity of carbon stereocenter. However, the authors did not specify how the diastereomeric ratios were determined—whether by NMR, GC, or another method. Besides, could the diastereoisomers be separated by column chromatography? All of the above information should be clearly presented in the main text or Supporting Information.

Response: We sincerely thank this reviewer for pointing it out. The diastereomeric ratios (dr) were determined by chiral HPLC analysis. HPLC analysis data are in the Supplementary Information. The diastereoisomers could not be separated by column chromatography. Please see our revised manuscript and Supporting Information.

(2) The authors did not provide any optimization of the reaction conditions in the Supporting Information. Including detailed reaction conditions is essential to help readers better understand the development and reproducibility of the method.

Response: We sincerely appreciate the reviewers' valuable and constructive comments. In response, we have provided comprehensive experimental data for the condition screening studies in the Supporting Information. Please see our revised the Supporting Information.

(3) In Table 1, the dr values should also be provided, and what is the structure of Ar₃ in ligand screening?

Response: We sincerely thank this reviewer for pointing it out. The dr values have been provided, and the structure of Ar₃ in ligand (3,5-Me₂-C₆H₃) was already shown in Table 1. Please see our revised manuscript.

(4) What is the origin of diastereoselectivity in the cases involving unsymmetrical diaryl diazomethanes? According to the DFT calculation results, it is likely that the $\pi\cdots\pi$ interaction between the aromatic group in the symmetric diazo compound and the aromatic group in amine borane, is the key factor leading to stereoselectivity tendency in the reaction. Detailed DFT calculation is suggested to give a clearer explanation for the unsymmetrical diaryl diazomethanes.

Response: We thank the reviewer for the valuable suggestions. To investigate the origin of diastereoselectivity for the unsymmetrical diaryl diazomethanes, we performed DFT calculations using diaryldiazomethane with one para-cyano-substituent as the substrate and **L3** as the model ligand in searching for the four stereoisomeric transition states. The results suggest that with unsymmetrical diaryldiazomethanes, the previously lowest-energy R-configured transition state underwent a conformational shift, rendering the **TS-R-R** (boron stereochemistry first, then carbon) configuration energetically most favorable due to an N-H $\cdots\pi$ interaction between the N-H bond of substrate **1a** and the phenyl ring, alongside a $\pi\cdots\pi$ interaction between the additional cyano-substituted phenyl ring and the catalyst's pendant phenyl ring (see Figure below). Although both weak interactions were present in **TS-R-S**, the N-H bond of **1a** preferentially engaged with the electron-rich aromatic ring, elucidating why larger electronic differences between the two aryl rings in asymmetric diaryldiazomethanes lead to higher dr values, such as **3o**.

Figure 1. DFT-calculated key transition state structures for unsymmetrical diaryldiazomethanes.

To better quantify the stereoselectivity in the transition states, we performed distortion-interaction analysis on the aforementioned four transition states, with the results summarized in the table below; **TS-R-R** exhibited the largest interaction energy, validating the strongest non-covalent interactions present, while the higher distortion energy of **TS-R-S** relative to **TS-R-R** indicated more pronounced

steric repulsion leading to its elevated energy, which corresponds precisely with the experimental findings (**3af-3aj**) showing that reducing the steric bulk of the N-substituent significantly decreases the dr value; this DI analysis further confirms that both weak interactions between substrates and spatial positioning effects are crucial for favoring the formation of the *R,R*-configured product. These new results were also added to the revised supplementary information.

Table 1. The DI analysis results of the four transition states.

Energy/ kcal mol ⁻¹	TS- R-R	TS- S-S	TS- R-S	TS- S-R
$\Delta E_{\text{dist}}(\text{carbene})$	5.3	1.9	6.6	2.7
$\Delta E_{\text{dist}}(\text{borane})$	0.5	0.5	0.7	1.1
$\Delta E_{\text{dist}}(\text{total})$	5.8	2.4	7.3	3.8
ΔE_{int}	-12.0	-10.4	-11.8	-11.8

(5) In Figure 5b, the S factor and C values should also be provided.

Response: We sincerely thank this reviewer for pointing it out. The S factor and C values have been provided. Please see our revised manuscript.

(6) In Figure 5d, **1a-1** was used as a chiral catalyst; however, the hydrogenated product was obtained in only 35% yield with 17% ee. Further optimization is clearly needed.

Response: We sincerely thank this reviewer for pointing it out. Optimization studies revealed that catalytic amounts of borane **1a-1** proved insufficient for achieving satisfactory reaction yields or enantioselectivity. In contrast, employing stoichiometric amounts of borane **1a-1** enabled reduction of both ketones and imines, with the cyclic imine **5b** exhibiting moderate enantiocontrol. These findings demonstrate the promising potential of amine boranes with a *N*-stereogenic center as a novel class of chiral transfer hydrogenation reagent. Please see Fig. 5d in our revised manuscript.

Reviewer #2:

Asymmetric synthesis of chiral molecules has long been a significant challenge in organic chemistry. Over the past few decades, numerous approaches have been explored for constructing C-centered chirality, while precise modulation of contiguous heteroatomic stereocenters, particularly those involving nitrogen and boron, remains complex. In this context, Song and coworkers have presented a robust protocol for constructing troublesome contiguous three-atomic (B, N, C) stereocenters, vicinal two-heteroatomic (B and N) stereocenters, and a nitrogen stereocenter via Cu/BOX-catalyzed asymmetric B-H insertion. A wide range of amine boranes and diazo compounds demonstrated high enantio- and diastereoselectivities.

Mechanistic investigations suggest that the reaction may undergo a kinetic resolution of racemic amine boranes, resulting in the simultaneous synthesis of both chiral unreacted substrates with a

single nitrogen stereogenic center and compounds featuring contiguous heteroatomic stereogenic centers. The enantioriched starting materials also show potential as chiral organocatalysts in the asymmetric hydrogenation of naphthyl ketones, albeit with inferior enantiocontrol. Additionally, DFT calculations provided further insights into the stereocontrol model. These results are intriguing, and I recommend publication after major revisions to address several deficiencies: **Response:** We sincerely thank this reviewer for the favorable comments on our work, we really appreciate it. We have provided detailed responses to your constructive suggestions point by point.

1) The B - H bond is retained in the enantioriched product. Is it possible for this compound to undergo Cu-catalyzed B - H insertion again, potentially assisted by **Int-L1** with the opposite configuration or a racemic BOX ligand?

Response: We sincerely appreciate the reviewers' valuable and constructive comments. In response, we did not observe the product of secondary insertion by using ligands with opposite configurations or racemic ligands.

2) Could the enantioriched product be utilized in an enantioselective hydrogenation, as illustrated in Fig. 5d? Would this provide superior enantiocontrol?

Response: We sincerely thank this reviewer for pointing it out. The enantioriched product, such as **3a**, cannot be used for enantioselective hydrogenation. Optimization studies revealed that catalytic amounts of borane **1a-1** proved insufficient for achieving satisfactory reaction yields or enantioselectivity. In contrast, employing stoichiometric amounts of borane **1a-1** enabled reduction of both ketones and imines, with the cyclic imine **5b** exhibiting moderate enantiocontrol. These findings demonstrate the promising potential of amine boranes with a N-stereogenic center as a novel class of chiral transfer hydrogenation reagents. Please see Fig. 5d in our revised manuscript.

3) There are several typographical errors in the manuscript that should be carefully re-checked. Here are some examples:

- In the abstract, redundant hyphens have been included, such as in “proto-col,” “trouble-some,” and “path-way.”
- On page 2, line 11, the “N” in “N-oxides” should be italicized.
- On page 2, line 20, a redundant space appears in “carbon- boron-nitrogen.”
- In Figure 5b, the first line “3a Yield (%)” appears awkwardly formatted.

Response: We sincerely appreciate the reviewers' valuable and constructive comments. The above problems have been revised. Additionally, we have conducted a comprehensive review of the full text. Please see our revised manuscript.

Reviewer #3:

The authors disclose an asymmetric B – H insertion reaction of diaryl diazo compounds by reacting the diazo compounds with cyclic boranes harboring an adjacent amine group. While the reaction is interesting, I am not convinced that it has enough conceptual novelty and significance to merit publication in Nat. Comm. The obtained products are highly specialized and do not provide a general access to a wide variety of B, N containing compounds.

Response: We sincerely thank you for your careful review of our manuscript and constructive suggestions, as well as for your serious research and profound insights in this field. We are deeply aware that your opinions are crucial for improving the quality of the manuscript. We would like to address the reviewers' concerns while explaining why and how our research will contribute to the existing knowledge and advances in the field, which we believe makes our entitled manuscript meet the high publication standards of *Nature Communications*.

1. Chiral molecules play important roles in various fields. In fact, in the family of heteroatomic stereocenters, the construction of nitrogen stereocenter is very challenging. Chiral nitrogen-stereogenic centers could be found in some natural products, drug molecules as well as chiral catalysts. Especially, the Feng's ligands containing nitrogen-containing stereocenters provide an effective approach for the synthesis of chiral compounds in the field of asymmetric catalysis (*Acc. Chem. Res.* **2011**, *44*, 8, 574 – 587). However, the construction of nitrogen-containing stereocentered compounds is still in its infancy, and the asymmetric catalytic strategy for constructing nitrogen-containing stereocentered compounds is extremely rare (*Tetrahedron Lett.* **2007**, *48*, 4183-4185; *Helv. Chim. Acta.* **2009**, *92*, 677-688; *Chem. Commun.* **1999**, *18*, 1787 – 1788; *Nature* **2021**, *597*, 70-76. *Angew. Chem. Int. Ed.* **2024**, *63*, e202404979. *Nat. Commun.* **2024**, *15*, 7317.)

2. The creation of novel compounds serves as a cornerstone of pharmaceutical research while also catalyzing scientific and industrial progress. Our synthetic protocol can not only prepare chiral amine boranes, but also provides a general platform for the construction of both vicinal nitrogen & boron stereogenic centers and nitrogen, boron & carbon continuous stereogenic frameworks, which has not been reported in literature, therefore, we believe that our findings would be of great interest to a very broad range of synthetic chemists, medical researchers, biochemists and other related researchers.

We sincerely appreciate the time and effort invested by this viewer in evaluating our manuscript. We look forward to any additional feedback or suggestions.

i) Authors have already reported a conceptually similar copper catalyzed B-H insertion reaction of alpha-diazo phosphonates (*Chemical Science*, **2024**, *15*, 7130-7135). Furthermore, several catalytic B-H insertion reactions have been reported on other systems (reference 43-54).

Response: We sincerely thank the reviewer for raising these important points. We sincerely apologize for any misunderstanding that may have arisen due to our inadequate explanation in the original manuscript. This work represents a significant departure from our previous work (*Chem. Sci.* **2024**, *15*, 7130-7135). While our previous work focused on the synthesis of chiral α -boryl

phosphate compounds, where the boron and phosphorus atoms were not directly connected, this work introduces the construction of enantiopure boron-coordinated N-centered compounds, vicinal N-B stereogenic centers and N-B-C continuous stereogenic frameworks. Additionally, previous catalytic B-H insertion reactions have mainly focused on the construction of C-centered chirality and relied on Rh catalysts (references 43-54).

ii) If the novelty of the current work is on generating chiral nitrogen center adjacent to chiral borane, then applications of obtained products to construct biologically/medicinally important chiral B- or N-containing compounds should have been given. In the absence of any application of borane group, it is not clear how the presence of chiral borane atom adjacent to nitrogen (Fig. 2-4) would allow construction of valuable chiral amines.

Response: We sincerely thank this reviewer for pointing this out. The nitrogen-stereogenic compounds depicted in Fig. 2-4 represent a novel class of chiral molecules, whose development for biological/medicinal applications constitutes a primary research focus of our group. Our preliminary investigations have revealed that amine borane exhibits promising characteristics as a potential chiral transfer hydrogenation reagent. For example, amine borane **1a-1** reduced cyclic imine **5b** to obtain moderate enantiocontrol. Please see Fig. 5d in our revised manuscript.

Reviewers' comments

Reviewer #1 (Remarks to the Author):

The authors have fully addressed all my previous concerns, and I recommend this manuscript for publication in Nature Communications without further revision.

Our Response: We sincerely thank this reviewer for his/her favorable comments on our manuscript, we really appreciate it!

Reviewer #2 (Remarks to the Author):

The authors have properly revised the manuscript and addressed the concerns raised by the reviewers. The publication is recommended.

Our Response: We sincerely thank this reviewer for his/her favorable comments on our manuscript, we really appreciate it!

Reviewer #3 (Remarks to the Author):

The authors have provided additional data (Fig. 5d) to demonstrate the utility of their new compounds. Also, the relative importance and novelty of this work in comparison to their previous work has been clarified. The revised manuscript could be accepted for publication after the authors add this clarification so that this work can be adequately placed in context to the prior art.

Our Response: We sincerely thank this reviewer for his/her favorable comments on our manuscript, we really appreciate it! The relative importance and novelty of this work have been updated in the introduction section, please see our revised manuscript.